# Corrosion Behavior of Alkyd-Resin-Coated Carbon Steel under Cathodic Polarization in Both Static and Flowing Seawater

**Hui Guo [1], Kun Zhou [2], Zhenliang Feng [1], Chengjie Li [3], Jie Xie [1], Jiyuan Ma [1], Xinyue Zhang [1], Xiaohui Wang [2], Kunshan Xu [1], Chuanpeng Li [4],\* and Jie Liu [1],\***

1  College of Chemistry and Chemical Engineering, Yantai University, Yantai 264005, China
2  Southwest Technology and Engineering Research Institute, Chongqing 400039, China
3  Shandong Engineering Research Center of Green and High-Value Marine Fine Chemical, Weifang University of Science and Technology, Shouguang 262700, China
4  Shandong Laboratory of Yantai Advanced Materials and Green Manufacturing, Yantai 264006, China
\*  Correspondence: lcp1102402081@163.com (C.L.); liujie6573@163.com (J.L.)

**Abstract:** The effect of cathodic polarization on the corrosion behavior of alkyd-resin-coated carbon steel with an artificial coating defect was researched using a wire beam electrode (WBE) and electrochemical impedance spectroscopy (EIS) in both static and flowing simulated solutions. The microscopic morphology and chemical structure of the organic coating were characterized by scanning electron microscopy (SEM) and infrared spectroscopy (FT-IR) to reveal the degradation mechanisms of organic coatings under different polarization potentials. The study found that the failure process of the alkyd coating could be accelerated by cathodic polarization. After 312 h of immersion, the impedance under −1100 mV was one order of magnitude lower than that under the open-circuit potential (OCP). The coating delamination became serious with the negative shifting of polarization potential, and the delamination area ratio under −1100 mV in both static and flowing seawater rose to 23% and 14%, respectively. Interestingly, the flowing condition of the immersion solution that combined with cathodic polarization exhibited a synergistic effect, which could accelerate (in the earlier stage) and then alleviate the delamination of the coating. Furthermore, the results showed that both the diffusion of the corrosion particles and the anodic dissolution reaction of the metal could be significantly affected by cathodic polarization and the flowing condition of the solution, which provides a possible approach to gain insight into the delamination of organic coating.

**Keywords:** wire beam electrode; cathodic polarization; coating delamination; static seawater; flowing seawater



## 1. Introduction

Without effective protection, the corrosion of metal is inevitable, which seriously affects the service life of equipment. Organic coatings or composite materials have been used to mitigate corrosion or damage to metal [1,2]. However, the marine environment is harsh and complex, and a single protection method can no longer meet the corrosion protection needs of marine equipment. The combined application of organic coatings and cathodic protection (CP) technology is currently the most economical and effective method to control the corrosion of metal structures such as bridges, ships and pipelines [3–5]. In the earlier stages of the double-protection system, the isolation function of organic coatings plays a key role as the corrosive solution has not permeated through the coating and the drop of the potential mainly occurs due to the higher coating resistance. Once the corrosive solution reaches the metal matrix, the coating defects appear. The corrosion of the exposed metal of the coating defects can be avoided, which mainly relies on cathodic protection [6]. In addition to the cathodic current, the cathodic delamination process can also be affected by other factors, such as seawater velocity, film thickness, cation mobility, temperature and potential [7–10].

In recent decades, much work has been devoted to the mechanisms of the cathodic delamination of coatings in seawater environments [11–14]. It is generally believed that the cathodic delamination of coatings is related to high alkalinity (pH value $\geq$ 14) at the coating/metal interface induced by the cathodic reduction reaction [15]. Experimental and theoretical studies have shown that the reported mechanisms of the cathodic delamination of the coated metal matrix mainly relates to the breaking of the bond between the coating and metal, the saponification of the polymer coating, and the displacement of the coating occurring at the coating/metal interface [16–18]. In addition, the short-lived and highly active intermediates (superoxide or hydroxyl radicals) of the oxygen reduction reaction are also considered to be responsible for coating delamination [19]. Sykes et al. [20] used a scanning kelvin probe (SKP) and scanning acoustic microscopy (SAM) to observe the cathodic delamination of the coating on a metal surface. They found that the coating delamination rate was influenced by the ion transport rate at the coating/metal interface. Sørensen et al. [21,22] studied the influence of environmental factors on the cathodic delamination of a coating in seawater, and they found that the decisive velocity step of cathode delamination was the transport of sodium ions from the defects to the cathode region. Bi et al. [23] confirmed that the cathodic delamination rate of epoxy resin could be accelerated by the increasing concentration of NaCl and the negative shifting of potential. Although the cathodic delamination mechanisms of organic coatings have been discussed in depth, the effects of cathodic polarization on the corrosion behavior of a metal matrix have not been studied in detail, and the degradation mechanism of a coating under cathodic polarization needs to be further discussed. Furthermore, the influence of flowing seawater on the metal corrosion process has not been considered either.

In fact, marine equipment can be greatly affected by the flowing conditions of seawater in real application processes; the diffusion of corrosive particles and the failure of organic coatings can be significantly accelerated by flowing seawater. In addition to the computational simulation [24], the effect of one or several factors on the failure behavior of the coating/metal system can be effectively investigated through the simulated seawater immersion test. Previously, we investigated the corrosion behavior of a coated carbon steel/copper alloy in both static and flowing simulated seawater environments [25]. We found that more severe delamination of the WBE occurred in flowing seawater, and the polarity conversion of the carbon steel was initiated by the anodic dissolution reaction. So far, however, in addition to our recent research, few works have been carried out related to the corrosion behavior of coated metal involving both the flow state and cathodic polarization. In the present work, the effect of cathodic polarization on the corrosion behavior of alkyd-resin-coated carbon steel with an artificial defect was investigated based on a WBE and EIS in both static and flowing simulated seawater environments. The induction of the cathodic delamination mechanism and the development process of the organic coating under different cathodic polarization potentials and seawater environments were studied in terms of current density distribution, EIS characteristics and macroscopic morphology. Furthermore, the microscopic morphology and the composition of the organic coating in contact with the metal matrix were also characterized by SEM and FT-IR to understand the degradation mechanism of the coating at the delamination areas. The results of the study could provide a theoretical basis for the maintenance of marine equipment and the development of anti-corrosion measures.

## 2. Experimental

### 2.1. Electrode and Solution Preparation

The WBE was made of 100 Q235 carbon steel wires with a diameter of 1.5 mm [26,27]. Table 1 shows the main components of the Q235 carbon steel obtained from the supplier. The steel wires with attached conductors were placed into a PVC pipe and a 10 × 10 matrix template, and a pitch of 0.5 mm was used to fix. Epoxy resin was filled into the PVC pipe to achieve the electrical insulation of the adjacent steel wires [28–30]. The total working area of the WBE was approximately 1.77 cm$^2$. As can be seen in Figure 1a, the surface of

the WBE was sanded with #400 and #800 silicon carbide sandpapers; deionized water and ethanol were used for cleaning and degreasing. The alkyd resin (Liangshan Jinchong Paint Industry Inc., Jining, China) was artificially applied to the surface of the WBE. The pencil hardness of the film was determined by pushing a pencil of gradually increasing hardness onto the film (ISO 15184-2012 Paints and varnishes-Determination of film hardness by pencil test). After standing at room temperature (25 °C) for seven days, the pencil hardness value of the alkyd resin coating was HB, indicating the coating was completely cured. The coating thickness (80 ± 5 μm) was measured by the AR-932 thickness gauge (Lianyungang Jinsheng Technology Inc., Lianyungang, China). The surface of the No. 45 electrode was artificially damaged; the area of the coating defect accounted for about 1% of the test area, as shown in Figure 1b. A 3.5 wt.% NaCl solution was used to simulate seawater, and the flowing condition was achieved by a Cantilever electric stirrer (Changzhou Yineng Experimental Instrument Factory, Changzhou, China). The stirring speed was 120 r/min.

**Table 1.** Chemical composition of Q235 carbon steel (mass fraction/%).

| Chemical Composition | C | Si | Mn | P | S | Fe |
|---|---|---|---|---|---|---|
| Q235 carbon steel | 0.18 | 0.30 | 0.32 | 0.04 | 0.03 | Balance |

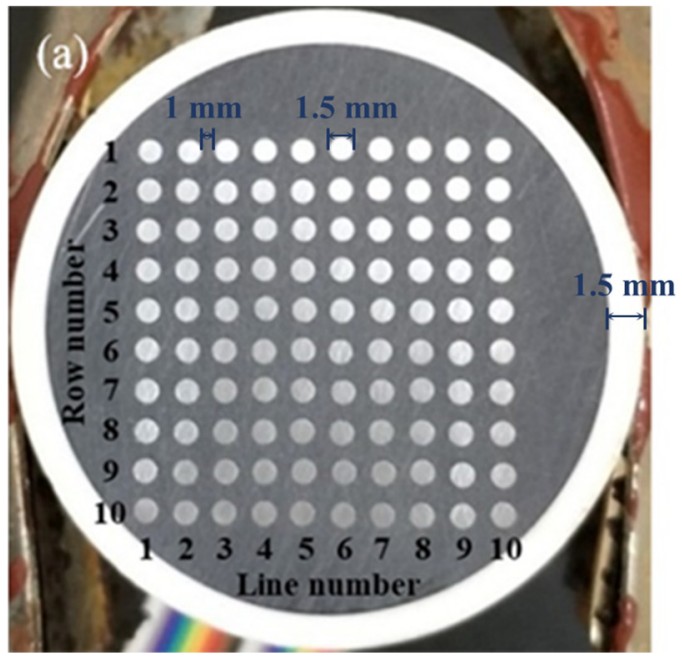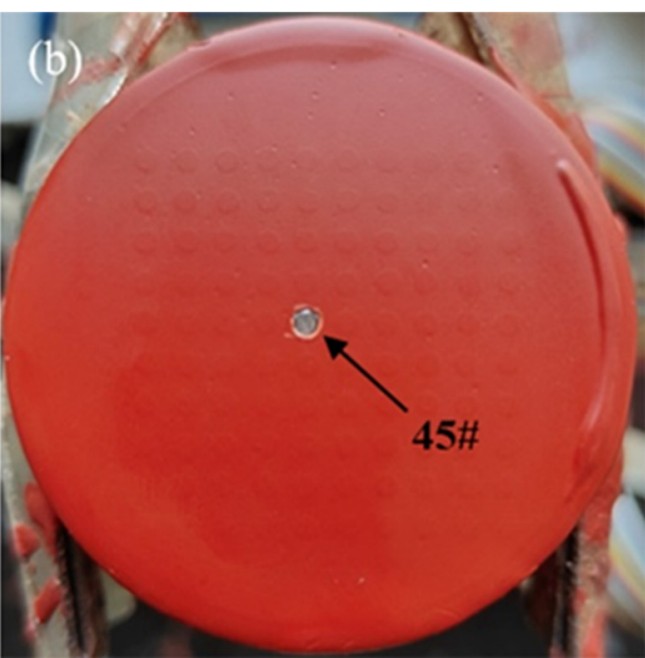

**Figure 1.** Pictures of the initial WBE (**a**) and the coated WBE with an artificial defect (**b**).

*2.2. Cathodic Polarization*

The prepared coating samples were immersed in the two kinds of simulated seawater environments. The cathodic polarization current was applied to the WBE using a potentiostat (DJS-292B, Shanghai Xinrui Inc., Shanghai, China). The working electrodes were all 100 of the electrodes, the reference electrode was a saturated calomel electrode (SCE), and the auxiliary anode was a high-purity graphite electrode. Three different cathodic polarization potentials—open-circuit potential (OCP), −850 mV $_{vs. SCE}$ and −1100 mV $_{vs. SCE}$—were applied to three different coated samples in separated electrolytic cells, respectively. The fluctuation range of the OCP values is given in Tables 2 and 3. Both the experimental test and the immersion process were conducted at room temperature.

**Table 2.** Fitting results of the EIS data under different immersion times in static seawater.

| Potential | Time (h) | $R_c$ ($\Omega \cdot cm^2$) | CPE1 ($S \cdot s^{n1} \cdot cm^{-2}$) | $n_1$ | $R_{ct}$ ($\Omega \cdot cm^2$) | CPE2 ($S \cdot s^{n2} \cdot cm^{-2}$) | $n_2$ | W ($S \cdot s^{0.5} \cdot cm^{-2}$) | Chi-Squared Value | Sum-of-Squares Value |
|---|---|---|---|---|---|---|---|---|---|---|
| OCP (−214~−216 mV) | 0.5 | $1.89 \times 10^9$ | $1.43 \times 10^{-9}$ | 0.95 | - | - | - | - | $1.88 \times 10^{-3}$ | $1.39 \times 10^2$ |
| | 48 | $1.61 \times 10^7$ | $2.59 \times 10^{-9}$ | 0.90 | $2.88 \times 10^6$ | $5.43 \times 10^{-7}$ | 0.72 | - | $1.05 \times 10^{-3}$ | $1.18 \times 10^2$ |
| | 168 | $1.41 \times 10^4$ | $3.84 \times 10^{-9}$ | 0.85 | $5.06 \times 10^4$ | $6.42 \times 10^{-6}$ | 0.39 | $1.53 \times 10^{-4}$ | $6.17 \times 10^{-4}$ | $1.28 \times 10^1$ |
| | 312 | $3.72 \times 10^3$ | $3.90 \times 10^{-9}$ | 0.86 | $1.75 \times 10^4$ | $3.61 \times 10^{-5}$ | 0.41 | $3.59 \times 10^{-4}$ | $7.17 \times 10^{-4}$ | $1.20 \times 10^1$ |
| −850 mV vs. SCE | 0.5 | $1.82 \times 10^9$ | $1.45 \times 10^{-9}$ | 0.95 | - | - | - | - | $1.84 \times 10^{-3}$ | $7.02 \times 10^{-2}$ |
| | 48 | $2.21 \times 10^5$ | $1.77 \times 10^{-9}$ | 0.94 | $6.75 \times 10^4$ | $1.71 \times 10^{-5}$ | 0.45 | - | $7.13 \times 10^{-4}$ | $2.69 \times 10^1$ |
| | 168 | $3.09 \times 10^4$ | $4.14 \times 10^{-9}$ | 0.91 | $1.83 \times 10^4$ | $8.57 \times 10^{-5}$ | 0.40 | - | $5.77 \times 10^{-4}$ | $1.84 \times 10^1$ |
| | 312 | $1.23 \times 10^4$ | $5.48 \times 10^{-9}$ | 0.88 | $9.13 \times 10^3$ | $4.56 \times 10^{-4}$ | 0.35 | - | $3.52 \times 10^{-4}$ | $1.26 \times 10^1$ |
| −1100 mV vs. SCE | 0.5 | $1.47 \times 10^9$ | $8.24 \times 10^{-10}$ | 0.95 | - | - | - | - | $2.12 \times 10^{-3}$ | $2.23 \times 10^2$ |
| | 48 | $1.29 \times 10^4$ | $2.88 \times 10^{-9}$ | 0.91 | $2.53 \times 10^4$ | $1.40 \times 10^{-4}$ | 0.55 | - | $2.25 \times 10^{-4}$ | $1.97 \times 10^1$ |
| | 168 | $2.25 \times 10^3$ | $1.61 \times 10^{-8}$ | 0.80 | $9.24 \times 10^3$ | $2.24 \times 10^{-4}$ | 0.51 | - | $2.76 \times 10^{-4}$ | $1.89 \times 10^1$ |
| | 312 | $3.80 \times 10^2$ | $1.73 \times 10^{-8}$ | 0.80 | $8.34 \times 10^3$ | $3.08 \times 10^{-4}$ | 0.36 | - | $3.75 \times 10^{-4}$ | $2.56 \times 10^1$ |

**Table 3.** Fitting results of the EIS data under different immersion times in flowing seawater.

| Potential | Time (h) | $R_c$ ($\Omega \cdot cm^2$) | CPE1 ($S \cdot s^{n1} \cdot cm^{-2}$) | $n_1$ | $R_{ct}$ ($\Omega \cdot cm^2$) | CPE2 ($S \cdot s^{n2} \cdot cm^{-2}$) | $n_2$ | W ($S \cdot s^{0.5} \cdot cm^{-2}$) | Chi-Squared Value | Sum-of-Squares Value |
|---|---|---|---|---|---|---|---|---|---|---|
| OCP (−218~−223 mV) | 0.5 | $1.99 \times 10^9$ | $1.84 \times 10^{-9}$ | 0.83 | - | - | - | - | $1.02 \times 10^{-3}$ | $1.15 \times 10^2$ |
| | 48 | $1.75 \times 10^5$ | $2.49 \times 10^{-9}$ | 0.95 | $7.56 \times 10^6$ | $2.73 \times 10^{-8}$ | 0.50 | $1.44 \times 10^{-6}$ | $2.32 \times 10^{-3}$ | $3.42 \times 10^2$ |
| | 168 | $3.06 \times 10^4$ | $6.59 \times 10^{-9}$ | 0.94 | $7.27 \times 10^3$ | $1.75 \times 10^{-6}$ | 0.59 | $4.17 \times 10^{-4}$ | $6.38 \times 10^{-4}$ | $1.41 \times 10^2$ |
| | 312 | $1.32 \times 10^4$ | $3.51 \times 10^{-9}$ | 0.93 | $1.82 \times 10^3$ | $2.54 \times 10^{-5}$ | 0.63 | $7.16 \times 10^{-4}$ | $2.13 \times 10^{-4}$ | $1.25 \times 10^2$ |
| −850 mV vs. SCE | 0.5 | $2.10 \times 10^9$ | $1.38 \times 10^{-9}$ | 0.83 | - | - | - | - | $1.21 \times 10^{-3}$ | $7.58 \times 10^2$ |
| | 48 | $2.74 \times 10^4$ | $2.54 \times 10^{-9}$ | 0.92 | $3.56 \times 10^4$ | $3.67 \times 10^{-5}$ | 0.73 | $3.33 \times 10^{-4}$ | $2.47 \times 10^{-4}$ | $8.69 \times 10^2$ |
| | 168 | $1.88 \times 10^4$ | $2.51 \times 10^{-9}$ | 0.88 | $1.65 \times 10^4$ | $8.28 \times 10^{-5}$ | 0.62 | $3.84 \times 10^{-4}$ | $3.71 \times 10^{-4}$ | $1.19 \times 10^2$ |
| | 312 | $2.37 \times 10^4$ | $1.34 \times 10^{-8}$ | 0.85 | $1.74 \times 10^3$ | $5.47 \times 10^{-4}$ | 0.59 | $2.55 \times 10^{-3}$ | $6.12 \times 10^{-4}$ | $1.17 \times 10^2$ |
| −1100 mV vs. SCE | 0.5 | $1.86 \times 10^9$ | $2.06 \times 10^{-9}$ | 0.94 | - | - | - | - | $4.19 \times 10^{-5}$ | $1.06 \times 10^1$ |
| | 48 | $1.46 \times 10^4$ | $1.34 \times 10^{-8}$ | 0.94 | $1.04 \times 10^5$ | $1.11 \times 10^{-4}$ | 0.67 | - | $5.73 \times 10^{-4}$ | $3.26 \times 10^2$ |
| | 168 | $2.97 \times 10^3$ | $2.02 \times 10^{-8}$ | 0.87 | $3.70 \times 10^3$ | $4.41 \times 10^{-4}$ | 0.55 | - | $1.43 \times 10^{-4}$ | $1.30 \times 10^2$ |
| | 312 | $4.80 \times 10^3$ | $2.75 \times 10^{-8}$ | 0.83 | $1.16 \times 10^2$ | $8.44 \times 10^{-4}$ | 0.78 | $2.65 \times 10^{-4}$ | $1.39 \times 10^{-4}$ | $1.49 \times 10^1$ |

### 2.3. EIS Measurement

Before the measurement, the cathodic polarization of the WBE needed to be removed so that the EIS tests of the coating/metal system could be performed under its free corrosion potential. The electrochemical workstation CST310 (Wuhan Corrtest Inc., Wuhan, China) was used to perform the EIS tests at a frequency ranging from $10^5$ Hz to $10^{-2}$ Hz. The total number of points in the EIS measurement was 42, and the frequency mode was chosen as logarithmic. The amplitude of the sinusoidal disturbance was 20 mV to obtain stable data [31–33]. The electrolyte solution used in the tests was 3.5 wt.% NaCl solution. An SCE was used as the reference electrode and a platinum wire electrode was used as the counter electrode. During the EIS measurement, the No. 45 electrode of the WBE was disconnected. The remaining 99 electrodes were coupled as the working electrodes, and the total area was about 1.75 cm$^2$. After the measurement, the 100 electrodes were short-circuited to each other. The test results were analyzed by ZSimpWin 3.6 software (Ametek Inc., Philadelphia, PA, USA) and mapped by Origin 2018 software (OriginLab Inc., Northampton, MA, USA).

### 2.4. Current Density Test

A multi-channel, zero-resistance ammeter (CST520, Wuhan Corrtest Inc., Wuhan, China) was used to measure the current density distributions of the WBE. The cathodic polarization applied to the WBE was similarly removed before measuring the current density. The method of measuring the current density distributions of the WBE was as follows: The No. 1 electrode of the WBE was used as an independent electrode, and the remaining 99 electrodes as a whole. The current densities of the loop composed of the No. 1 electrode and all the remaining 99 electrodes were measured. After the measurement of the No. 1 electrode, it was reconnected to the WBE, and the same measurement was repeated for the No. 2 electrode until all 100 electrodes were measured.

### 2.5. Surface Morphology Analysis

Macroscopic morphological changes on the WBE surface were recorded using a phone (P40, Huawei Technologies Co. Ltd., Shenzhen, China). Microscopic morphology of the bottom surface of the coating in the cathodic delamination region was observed using a scanning electron microscope (Quanta 200, FEI Ltd., Hillsboro, OR, USA). A high-energy electron was used by SEM to observe the morphological structure of the sample surface at a high magnification [34]. Prior to the test, the surface of the coatings was sprayed with gold to make it conductive. The accelerating voltage used for the test was 20 kV.

### 2.6. FT-IR Analysis

The functional group composition of the coating in contact with the metal matrix at the delamination areas was characterized by an FT-IR spectrometer (PerkinElmer Inc., Waltham, MA, USA) [34]. The coating samples were scanned 32 times with a resolution of $4 \text{ cm}^{-1}$ in a range of $400–1600 \text{ cm}^{-1}$ to obtain the FT-IR spectrum.

## 3. Results and Discussion

### 3.1. Current Density Distributions of the WBE

Figure 2 shows the current density distributions of the WBE under the condition of OCP in the two seawater environments. As displayed in Figure 2(a$_1$), an anodic current peak with a current density of $3.34 \text{ µA·cm}^{-2}$ occurred at the No. 45 electrode when the WBE was immersed in static seawater for 48 h. Cathodic current peaks appeared on the surface of the No. 96 and No. 100 electrodes. This indicated that the anodic dissolution reaction occurred at the coating defect. Water and oxygen diffused to the surface of the electrodes through the micro-defects of the coating, which led to the cathodic reduction reaction at the coating/metal interface [35,36]. The No. 45 electrode served as an anode in the flowing seawater, and its current density was $5.40 \text{ µA·cm}^{-2}$ as shown in Figure 2(a$_2$). Cathodic current peaks appeared on the surface of the No. 54, No. 60, No. 70 and No. 71 electrodes. As shown in Figure 2(b$_1$,b$_2$), when the immersion time reached 312 h in flowing seawater, the No. 45 electrode had a higher anodic current density than in static seawater. Eight cathodic current peaks appeared in static seawater, while 14 cathodic current peaks appeared in flowing seawater. This meant that the electrochemical reaction on the WBE was more active in flowing seawater under the OCP condition. From the above results, during the whole immersion process, the No. 45 electrode always served as an anode to be corroded, while the electrodes in the other areas served as a cathode to be protected.

Figure 3 shows the current density distributions of the WBE under $-850 \text{ mV}_{\text{vs. SCE}}$ in the two seawater environments. When the immersion time lasted for 48 h, as illustrated in Figure 3(a$_1$), the current density range widened to $-16.13 \text{ µA·cm}^{-2}$ to $7.78 \text{ µA·cm}^{-2}$, which was significantly larger than the current density distribution interval, and the current density distribution characteristics of the WBE under the OCP condition were significantly different at the same immersion time. The reason was that the high oxygen concentration at the No. 45 electrode caused an oxygen reduction reaction, and the electrodes under the coating underwent a metal dissolution reaction due to the diffusion of corrosive particles. Simultaneously, the current density of the WBE in flowing seawater was in the range of $-19.83 \text{ µA·cm}^{-2}$ to $10.76 \text{ µA·cm}^{-2}$, which was higher than that in static seawater under the same conditions, indicating that the electrochemical reaction of the electrode surface in flowing seawater was more active at this time. It is noteworthy that the current density at electrode 45 in the two seawater environments was similar after 312 h of immersion, and the same numbers of anodic current peaks were found.

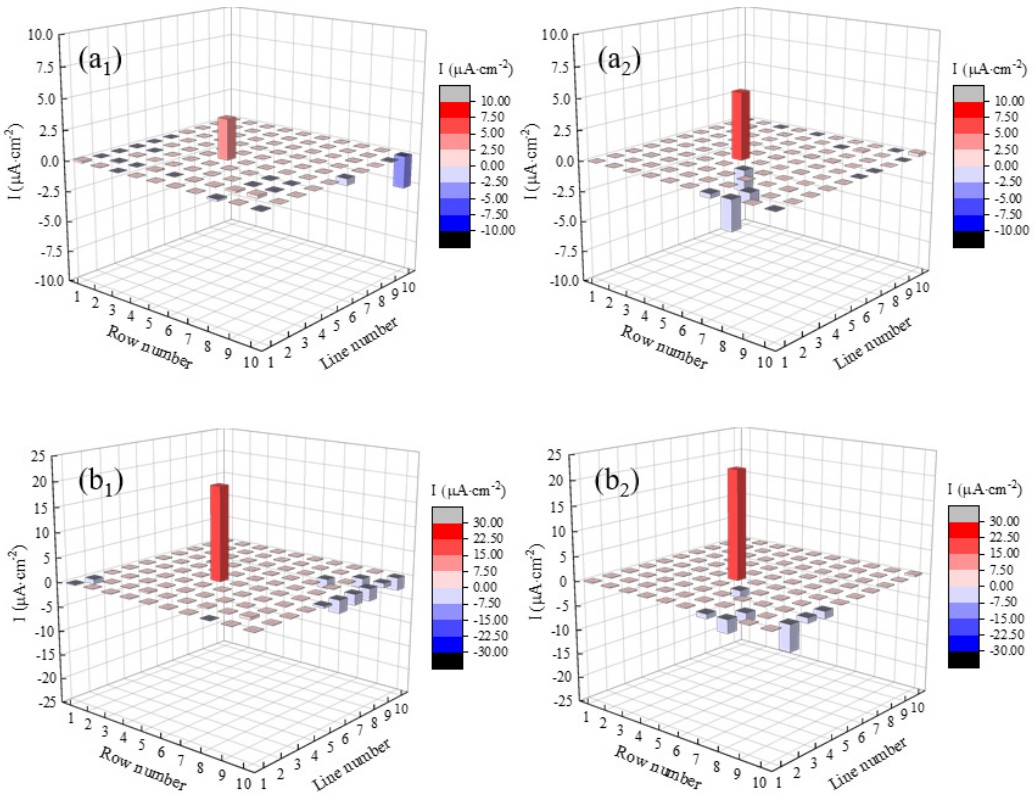

**Figure 2.** Current density distributions of the WBE under open-circuit potential in static seawater (**a₁**,**b₁**) and flowing seawater (**a₂**,**b₂**): (**a₁**,**a₂**) 48 h; (**b₁**,**b₂**) 312 h.

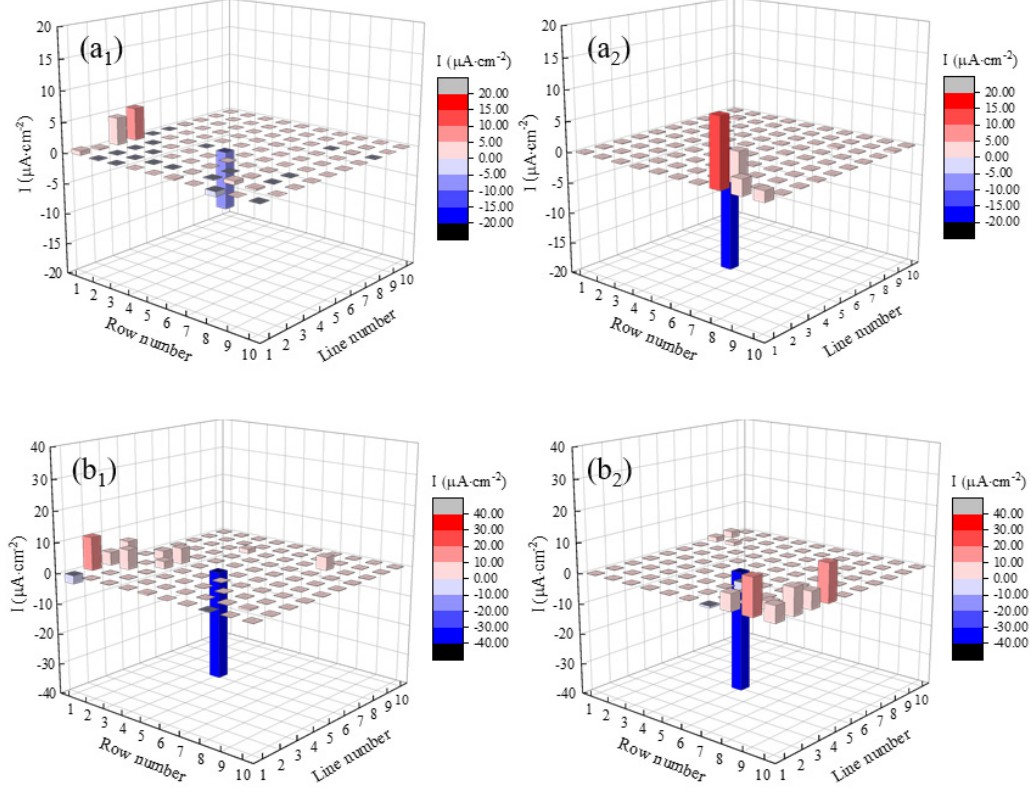

**Figure 3.** Current density distributions of the WBE under −850 mV vs. SCE immersed in static seawater (**a₁**,**b₁**) and flowing seawater (**a₂**,**b₂**): (**a₁**,**a₂**) 48 h; (**b₁**,**b₂**) 312 h.

Figure 4 shows the current density distributions of the WBE under $-1100$ mV $_{\text{vs. SCE}}$ in the two seawater environments. As shown in Figure 4($a_1$,$a_2$), when immersed for 48 h, there were four anodic current peaks in static seawater, and five anodic current peaks were found in flowing seawater. The current density soared to a range of $-52.51$ $\mu$A·cm$^{-2}$ to 25.01 $\mu$A·cm$^{-2}$, which was larger than that in static seawater. This indicated that the electrochemical reaction rate of the electrode in flowing seawater was also high in the first 48 h. The cathodic reaction at the coating/metal interface was dominated by hydrogen evolution, and the hydrogen cations could react with the metal oxide layer on the matrix and lead to oxide layer dissolution [37]. It is worth noting that when immersed for 312 h, there were 21 anodic current peaks in static seawater and only 14 anodic current peaks in flowing seawater. The current density values in both static and flowing seawater were 83.18 and 89.13 $\mu$A·cm$^{-2}$, respectively. The results showed that the electrochemical reaction area in flowing seawater became relatively small at this time. The specific reasons are explained in the following sections.

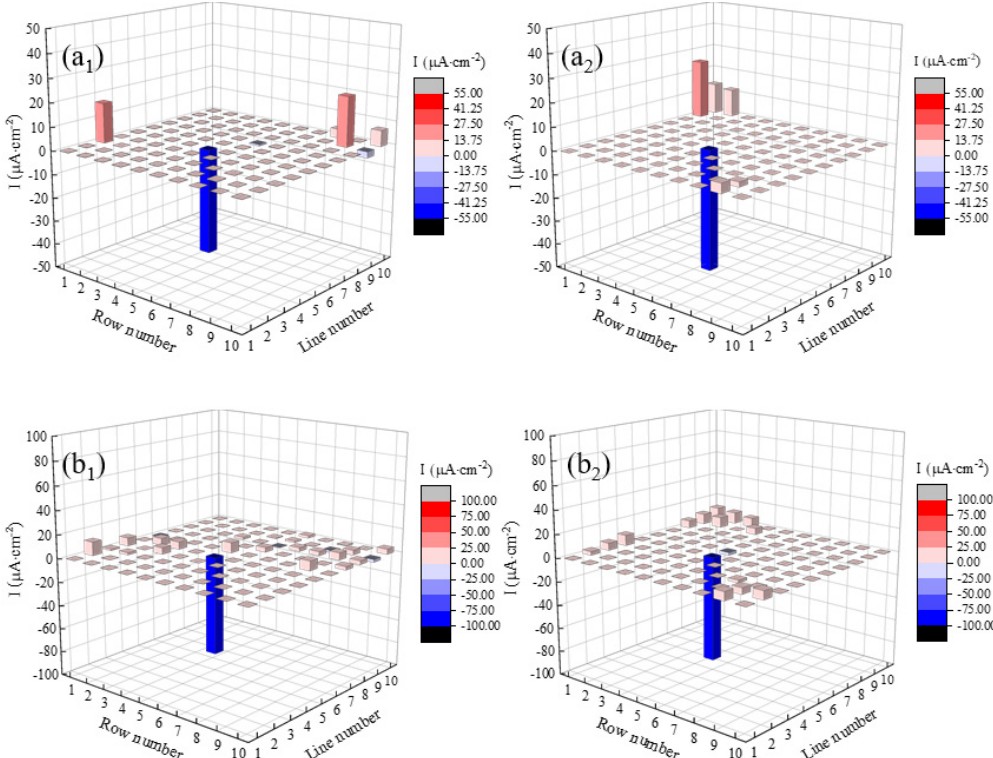

**Figure 4.** Current density distributions of the WBE under $-1100$ mV $_{\text{vs. SCE}}$ immersed in static seawater ($a_1$,$b_1$) and flowing seawater ($a_2$,$b_2$): ($a_1$,$a_2$) 48 h; ($b_1$,$b_2$) 312 h.

According to the above discussion, the increasing current densities of the WBE were related to more severe damage of the organic coating. Under the condition of OCP, the No. 45 electrode located at the coating defect served as an anode, the corrosion of which was accelerated. At the same time, the electrodes under the coating served as the cathode that was to be protected. Under cathodic polarization, the damage of the coating was more severe with the negative shifting of polarization potential, and the electrode at the coating defect was protected to a certain extent. The electrochemical reaction on the surface of the electrodes was more active in flowing seawater, which was mainly attributed to the increasing diffusion rate of corrosive particles in the flowing seawater condition [38]. Under the condition of $-1100$ mV $_{\text{vs. SCE}}$, the number of anodic current peaks was not significantly different in the two seawater environments in the first 47 h. When immersed for 312 h, the number of anodic current peaks in static seawater was more than that in flowing seawater. This indicated that the electrochemical reaction area on the surface of the WBE in flowing

seawater was small, which was related to the decreased delamination rate of the coating in flowing seawater.

### 3.2. EIS Characteristics of the WBE

Figures 5 and 6 present the EIS characteristic changes of the coated areas of the WBE with immersion time under different cathodic polarization potentials in the two seawater environments. As shown in Figure 5($a_1$) and Figure 6($a_1$), after being immersed for 0.5 h, each set of data points and the fitted curve in both static and flowing seawater individually displayed a large capacitive loop, and the impedance of the coated areas of the WBE reached $10^9$ $\Omega \cdot cm^2$, indicating that the coating was working well as a barrier [26,39,40]. At this time, the impedance spectrum fitted well with Circuit A in Figure 7a, where $R_s$ and $R_c$ are electrolyte resistance and coating resistance, respectively. Since the capacitive frequency response characteristics of solid electrodes did not coincide with the pure capacitance, there were always large or small deviations. To eliminate this "dispersion effect", the capacitance was represented by a constant phase element CPEx. CPE1 represented the coating capacitance. Tables 2 and 3 show the electrochemical parameters for fitting the EIS results to the two seawater environments, respectively. The Chi-squared and sum-of-squares values were used to test the fit of the data.

For 48 h of immersion, the Nyquist plots of the coated areas in static seawater are displayed in Figure 5($b_1$). Under the condition of OCP, it can be seen that two time constants were found and the impedance dropped to $1.01 \times 10^7$ $\Omega \cdot cm^2$. This means that corrosive particles diffused into the metal matrix through the micro-defects of the coating and the electrochemical reaction occurred at the coating/metal interface [41,42]. Therefore, Circuit B was chosen to fit the impedance spectrum data, in which the constant phase element CPE2 was used to represent the double-layer capacitance and $R_{ct}$ was the charge-transfer resistance. Under the two cathode polarization conditions, there were two obvious capacitive loops in the Nyquist plots. The impedance fitted by Circuit B in the low-frequency region were reduced to $6.82 \times 10^5$ $\Omega \cdot cm^2$ and $8.77 \times 10^4$ $\Omega \cdot cm^2$, respectively. This indicated that the protective performance of the coating decreased significantly under the condition of cathodic polarization. The Nyquist plots and fitting results for flowing seawater are shown in Figure 6($b_1$). The tail of the curve under the condition of OCP exhibited diffusion impedance characteristics, which indicated that the metal under the coating was relatively seriously corroded, and the corrosion products underwent a certain degree of accumulation at the coating/metal interface. The equivalent Circuit C was used to fit the impedance spectrum data. The Warburg impedance ($W$) showed that the accumulation of corrosion products formed a diffusion layer on the metal surface, and the corrosion reaction under the coating was controlled by the mass transfer diffusion process [35]. No diffusion tails appeared under the two polarization conditions, indicating that the metal matrix could be protected to a certain extent with the application of polarization potentials. At this point, the impedance values corresponding to the OCP, $-850$ mV $_{vs. SCE}$ and $-1100$ mV $_{vs. SCE}$ conditions were $8.89 \times 10^6$ $\Omega \cdot cm^2$, $5.01 \times 10^5$ $\Omega \cdot cm^2$ and $4.21 \times 10^4$ $\Omega \cdot cm^2$, respectively. It was shown that the impedance value decreased gradually with the increase in the cathodic potential, and the failure process of the coating was accelerated by cathodic polarization.

When the immersion time was increased to 168 h in static seawater, a diffusion tail appeared in the low-frequency region under the condition of OCP in Figure 5($c_1$), indicating that the electrochemical reaction on the surface of the WBE was affected by the diffusion process of the corrosion products [43,44]. Circuit C in Figure 7c was selected to fit the EIS data. It could be seen from the Nyquist plots under the condition of $-1100$ mV $_{vs. SCE}$ that the radius of curvature of the arc in the high-frequency region of the curve was significantly smaller than that in the low-frequency region. This indicated that the electrochemical process on the surface of the WBE dominated the contribution to the EIS data. As shown in Figure 6($c_1$), the impedance of the organic coating was further reduced under the conditions of polarization in flowing seawater. A diffusion tail appeared in the low-frequency region

when the polarization potential was $-850$ mV $_{vs. SCE}$, which indicated that the corrosion of the metal matrix was intensified. At this time, the impedance value dropped to $4.54 \times 10^4$ $\Omega \cdot cm^2$, indicating that the coating had almost lost its protective effect.

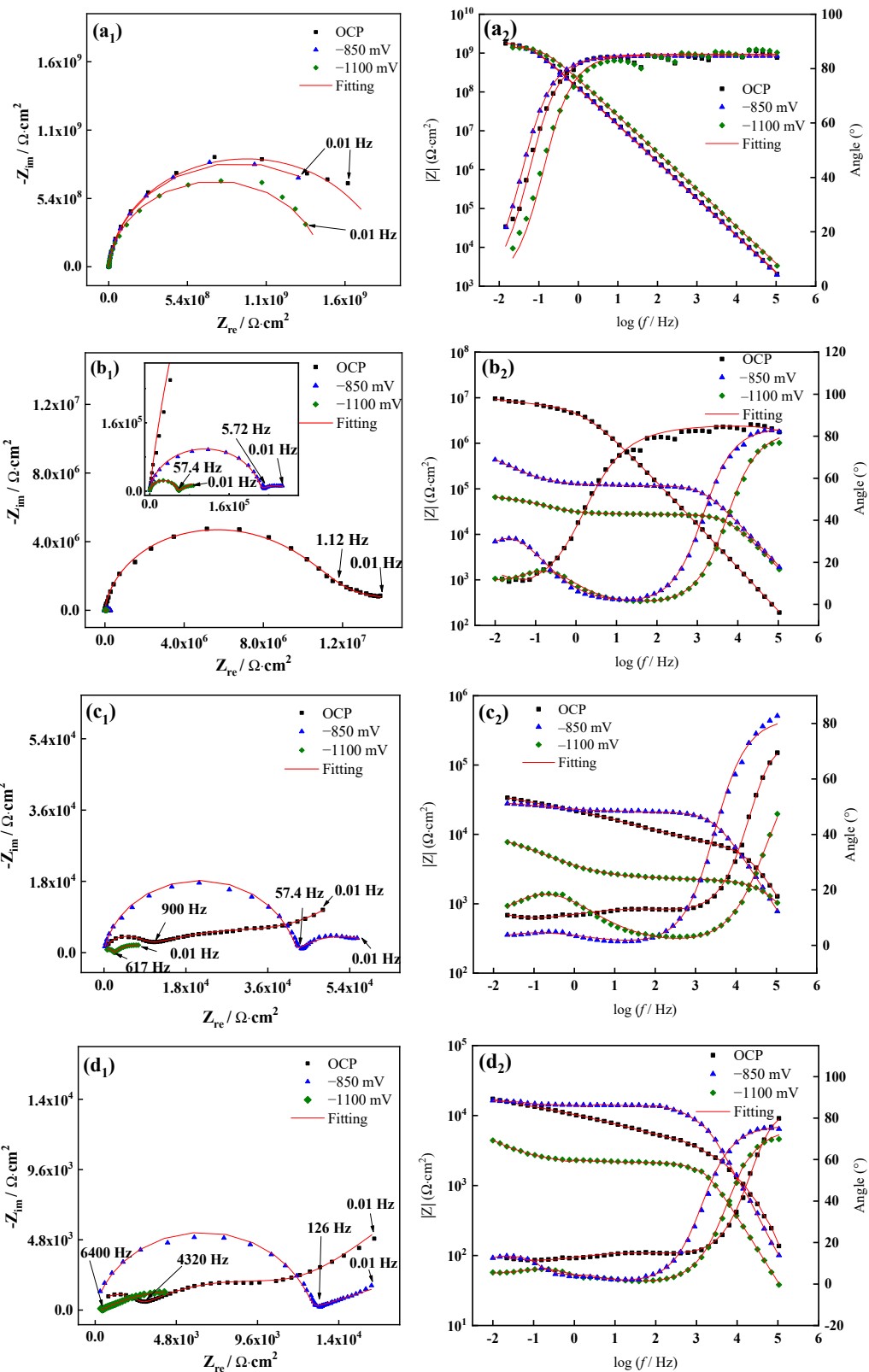

**Figure 5.** Nyquist plots ($a_1$–$d_1$) and Bode plots ($a_2$–$d_2$) of the WBE in static seawater at different immersion times: (**a**) 0.5 h; (**b**) 48 h; (**c**) 168 h; (**d**) 312 h.

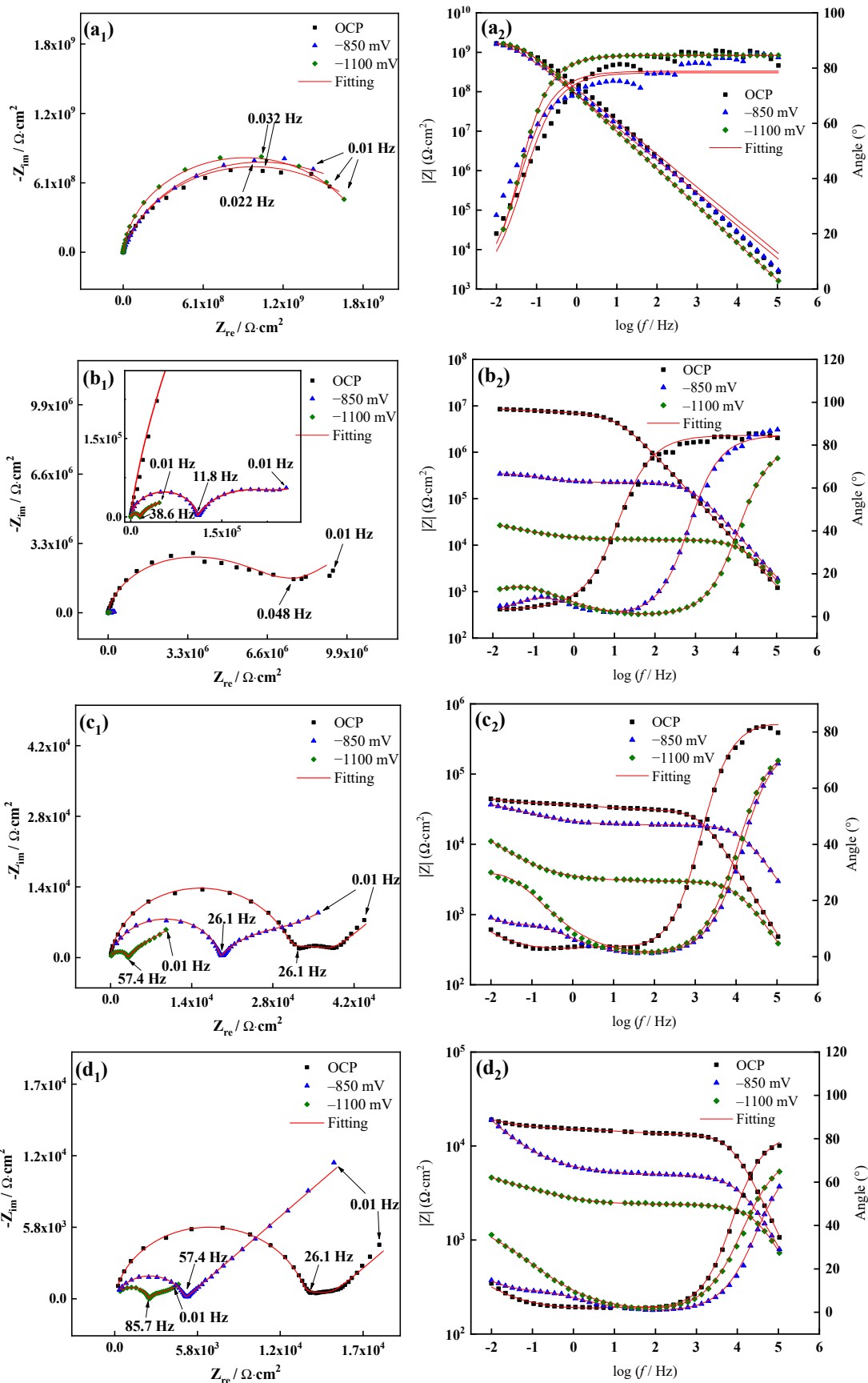

**Figure 6.** Nyquist plots (**a₁–d₁**) and Bode plots (**a₂–d₂**) of the WBE in flowing seawater at different immersion times: (**a**) 0.5 h; (**b**) 48 h; (**c**) 168 h; (**d**) 312 h.

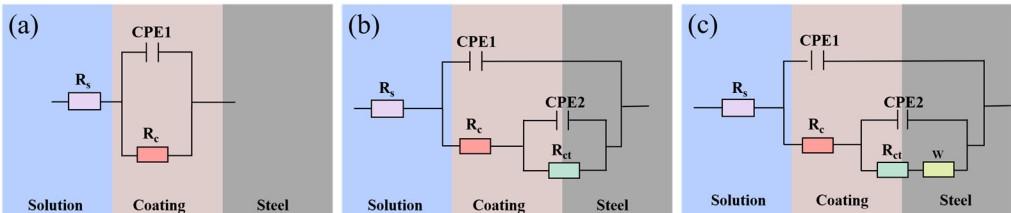

**Figure 7.** Equivalent circuits of organic coating systems after different immersion times. (**a**) R(CR); (**b**) R(C(R(CR))); (**c**) R(C(R(C(RW)))).

As shown in Figure 5($d_1$) and Figure 6($d_1$), when the test was carried out for 312 h, the impedance value under the condition of OCP in static seawater was $2.34 \times 10^4$ $\Omega \cdot cm^2$, which was smaller than that in flowing seawater ($2.79 \times 10^4$ $\Omega \cdot cm^2$). The reason was that the electrochemical reaction was more intensive in flowing seawater, more corrosion products accumulated on the surface of the WBE, and the transport of ions was hindered, which led to a higher impedance [44]. Under the condition of $-1100$ mV $_{vs.\ SCE}$, the EIS curve of the coating in static seawater had only two capacitive reactance arcs, indicating that the electrodes under the coating were well-protected. However, a diffusion tail was found in the EIS curve in flowing seawater. This indicated that the protective effect of cathodic polarization was weakened and the metal matrix was corroded.

Based on the above analysis, the coating resistance decreased significantly due to the application of the polarization potential, and this phenomenon became more obvious as the polarization potential became more negative. Under the OCP condition, the diffusion of corrosive particles (such as water and oxygen) into the coating was accelerated in flowing seawater, so the declination rate of the coating resistance $R_c$ was relatively accelerated, and the corrosion products accumulated much earlier at the coating/metal interface. Under the $-850$ mV $_{vs.\ SCE}$ condition, a diffusion tail was discovered earlier in flowing seawater. This showed that the protection effect of $-850$ mV $_{vs.\ SCE}$ on the metal matrix was relatively poor in flowing seawater. When the test was carried out for 168 h under the $-1100$ mV $_{vs.\ SCE}$ condition, there were no Warburg impedance characteristics in the EIS curves of the two seawater environments, which showed that the metal matrix was well-protected. When immersed for 312 h, a diffusion tail appeared in flowing seawater, indicating that the corrosion products had accumulated and the metal matrix was corroded. This showed that the electrodes under the coating were better protected at the more negative polarization potentials.

*3.3. Morphology of Organic Coating*

Figure 8 shows the morphology of the WBE when immersed for 312 h under different cathodic polarization potentials in the two seawater environments. The morphologies of the WBE in different seawater environments were compared. It can be seen from Figure 8a, under the OCP condition, that the surface of the No. 45 electrode was covered by yellow corrosion products in the two seawater environments. In addition, the corrosion products on the surface of the No. 45 electrode in flowing seawater were relatively porous and accompanied by a diffusion phenomenon. This indicated that the corrosion products had difficulty closely adhering to the No. 45 electrode in flowing seawater, which would further aggravate the corrosion [45]. Under the cathodic polarization conditions, as shown in Figure 8b,c, fewer yellow corrosion products appeared on the surface of the No. 45 electrode in static seawater. However, the No. 45 electrode was covered by dense black corrosion products in flowing seawater, showing the high corrosion degree of the No. 45 electrode in flowing seawater. This indicated that the protective effect of cathodic polarization on the metal at the coating defect was reduced due to the accelerated electrochemical reaction of the corrosion of the metal matrix in flowing seawater. It is worth noting that the number of blisters in flowing seawater was significantly less than that in static seawater under the condition of $-1100$ mV $_{vs.\ SCE}$. This was consistent with the conclusion drawn from

Figure 4, indicating that the WBE had a lower delamination rate in flowing seawater at this point.

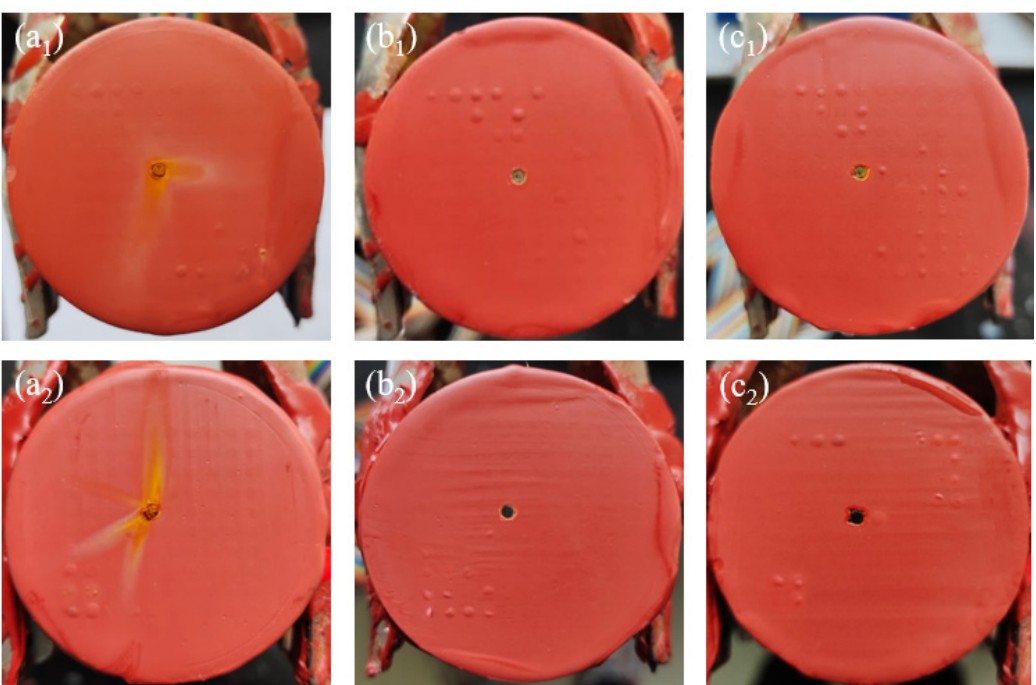

**Figure 8.** Morphology of the WBE under OCP (**a₁,a₂**), −850 mV vs. SCE (**b₁,b₂**) and −1100 mV vs. SCE (**c₁,c₂**) at 312 h of immersion time in static seawater (**a₁–c₁**) and flowing seawater (**a₂–c₂**).

The numbers of blisters under different polarization potentials were compared. The results showed that the number of blisters increased with the negative shifting of the potential under the same seawater condition, indicating a further increase in the degree of coating delamination. Under the condition of OCP, the defective area of the coating served as the anode area, and the metal under the coating served as the cathode to provide electrons for the oxygen reduction reaction. The accumulation of $OH^-$ generated by the cathodic reduction reaction at the coating/metal interface led to an increase in the alkalinity of the environment, which caused the partial coating delamination of a small area of electrodes. Under the condition of cathodic polarization, the anodic dissolution reaction of the metal was inhibited to a certain extent, and the following two electrochemical reactions mainly occurred:

$$O_2 + 2H_2O + 4e^- \rightarrow 4OH^- \tag{1}$$

$$2H_2O + 2e^- \rightarrow H_2 + 2OH^- \tag{2}$$

It was found that applying cathodic polarization to the coating/metal system could provide sufficient electrons for the two cathodic reactions, which increased the driving force for the cathodic delamination of the coating. The delamination degree of the coating increased with the negative shifting of the cathodic polarization potential under the same immersion time.

Based on the above discussion, the cathodic reaction and delamination rate at the coating/metal interface were increased with the negative shift of the cathodic potential. Additionally, the protective effect of cathodic polarization on the No.45 electrode was reduced due to the accelerated electrochemical reaction of the corrosion of the metal matrix in flowing seawater.

### 3.4. Development Process of Cathodic Delamination

In order to more intuitively understand the development process of the cathodic delamination of the coating, the changes in the delamination area of the coating were analyzed. Regarding the determination of the delamination area, in addition to the directly observable area, the area where the absolute value of the current density |I| was higher than 1 µA·cm$^{-2}$ was also a delamination area of the coating [46,47]. Therefore, the results of the current density distribution could be used to plot the delamination area of the WBE with immersion time, as shown in Figure 9. The delamination area ($A_d$) ratio of the coating could be expressed by the formula $A_d/A_{tol}$, in which the $A_{tol}$ was the total area of the coating on the electrodes.

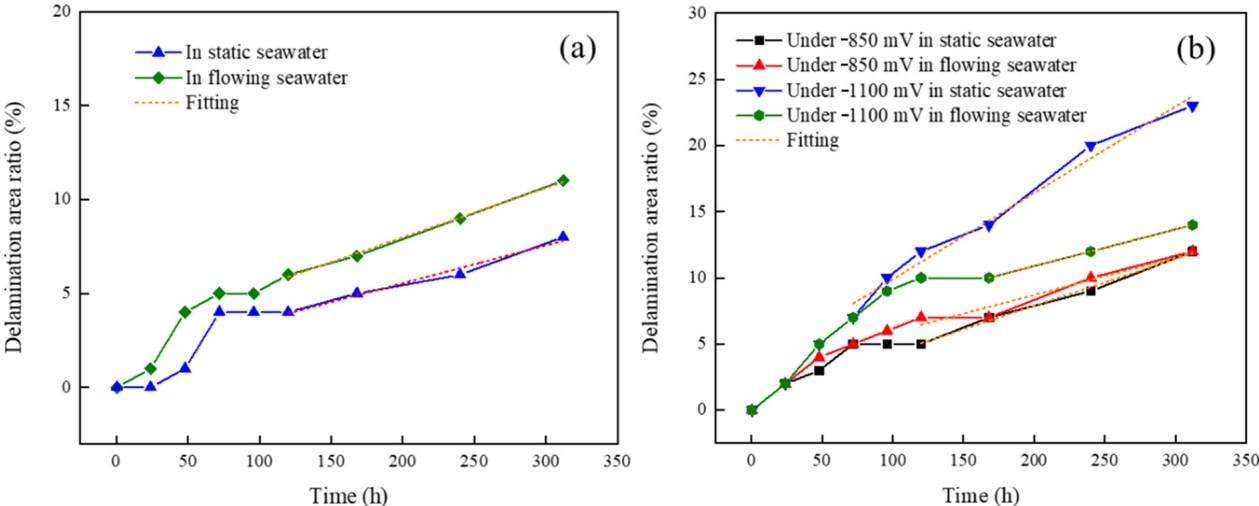

**Figure 9.** Evaluation of the delamination area of the coating under OCP (**a**), −850 mV $_{vs. SCE}$ and −1100 mV $_{vs. SCE}$ (**b**) in both static and flowing seawater with immersion time.

As shown in Figure 9a, the previous results were reconfirmed. Under the OCP condition, the coating consistently had a higher delamination rate in flowing seawater. The same variation trend of the delamination area of the coating was observed. The same linear growth trend was found after 120 h, indicating that the cathodic delamination mechanism of the coating was not affected by the seawater environments. Table 4 shows the linear fitting results of the curve. The slope of the corresponding curve in the linear increase stage of the coating delamination area in flowing seawater was 0.0264, which was higher than 0.0201 in static seawater. When immersed for 312 h, the delamination area ratio of the organic coating in static seawater was 8%, which increased to 11% in flowing seawater. This was mainly caused by the accelerated diffusion of corrosive particles in flowing seawater.

**Table 4.** Linear fitting results of the curves of the delamination area ratio under the condition of OCP versus time.

| Condition | Slope | Variance (%) |
|---|---|---|
| static seawater | 0.0201 | 98.0 |
| flowing seawater | 0.0264 | 99.5 |

Figure 9b shows the variation of the delamination area ratio of the WBE under cathode polarization. Under the condition of −850 mV $_{vs. SCE}$, the coating delamination rate in flowing seawater was not significantly different from that in static seawater during the period of immersion. The linear fitting results in Table 5 show that the slope of the curve in static seawater was 0.0354, and the slope of the curve in flowing seawater was 0.0283. This illustrated that the coating delamination rate in static seawater was slightly higher than

that in flowing seawater. Under the condition of $-1100$ mV $_{vs.\ SCE}$, the coating cathodic delamination rates in the two seawater environments were not significantly different for the first 72 h. Then, the cathode delamination rate in static seawater increased rapidly and exceeded that in flowing seawater. The slope of the curve in static seawater was 0.0278, and the slope of the curve in flowing seawater was 0.0653. This showed that with the increase in the delamination area of the coating, the area where the anodic dissolution reaction occurred on the electrode under the coating also increased. Due to the rapid diffusion of the corrosion particles in flowing seawater, a more serious anodic dissolution reaction occurred in flowing seawater. Under cathodic polarization, the cathodic polarization of the metal matrix was weakened, and the promotion effect of cathodic polarization in the cathodic reaction also decreased, resulting in a low cathodic delamination rate of the coating in flowing seawater.

**Table 5.** Linear fitting results of the curves of the delamination area ratio under the condition of cathodic polarization versus time.

| Potential | Condition | Slope | Variance (%) |
|---|---|---|---|
| $-850$ mV $_{vs.\ SCE}$ | static seawater | 0.0354 | 99.3 |
| | flowing seawater | 0.0283 | 94.4 |
| $-1100$ mV $_{vs.\ SCE}$ | static seawater | 0.0653 | 98.1 |
| | flowing seawater | 0.0278 | 100 |

*3.5. Morphology and Composition of the Coating in the Delamination Area*

Figure 10($a_1$,$b_1$,$c_1$) displays the morphologies of the coating in contact with the metal matrix under three cathodic polarization potentials for 312 h of immersion time in static seawater. Under the OCP condition, as shown in Figure 10($a_1$), a clear void appeared in the central region, and most of the region remained relatively smooth and uniform. As can be seen in Figure 10($b_1$), the coating surface became rough and uneven, and a deeper void was found under the cathodic polarization of $-850$ mV $_{vs.\ SCE}$. As displayed in Figure 10($c_1$), when the immersion time reached 312 h under the condition of $-1100$ mV $_{vs.\ SCE}$, a large number of voids were found on the surface of the coating, and the defects were largely spread throughout the coating. These phenomena proved that the degradation of the coating could be accelerated by cathodic polarization.

From Figure 10($a_2$,$b_2$,$c_2$), it can be seen that the defects of the coating morphology of the contact surface with the metal matrix under the three polarization potentials were more obvious than that in static seawater. A large number of voids and large-scale bumps were found on the coating surface in flowing seawater. Meanwhile, combined with the curve of the delamination area ratio of the coating, it could be seen that the cathodic delamination area of the coating was larger in static seawater under cathodic polarization, but the degradation of the alkyd resin in flowing seawater was more serious.

The FT-IR results of the coating (in contact with the metal matrix at the delamination areas) after 312 h of immersion are shown in Figure 11. The typical peaks of the alkyd resin were observed in the spectra. The broad band at 3415 cm$^{-1}$ was due to OH groups, the peaks at 2954 cm$^{-1}$ and 2870 cm$^{-1}$ represented C$-$H asymmetric and symmetric stretching vibrations of methylene groups, and the CH$_2$ bending vibration peak was observed at 1429 cm$^{-1}$. The peaks at 1635 cm$^{-1}$ and 1730 cm$^{-1}$ were assigned to the C=O stretching vibration of the carboxylic acids and esters, and the peak at 1153 cm$^{-1}$ was determined as the C$-$O$-$C stretching vibration [48–50]. When the coating/metal system under the conditions of $-850$ mV $_{vs.\ SCE}$ and $-1100$ mV $_{vs.\ SCE}$ were immersed for 312 h, it could be observed that the intensity of the OH-group peak around 3415 cm$^{-1}$ was significantly increased and broadened, and the intensity of the peak representing the carbonyl stretching vibration at 1635 cm$^{-1}$ also increased. The increase in these bands indicated that the organic coating degraded during the immersion process to form alcohol and carbonyl substances [50].

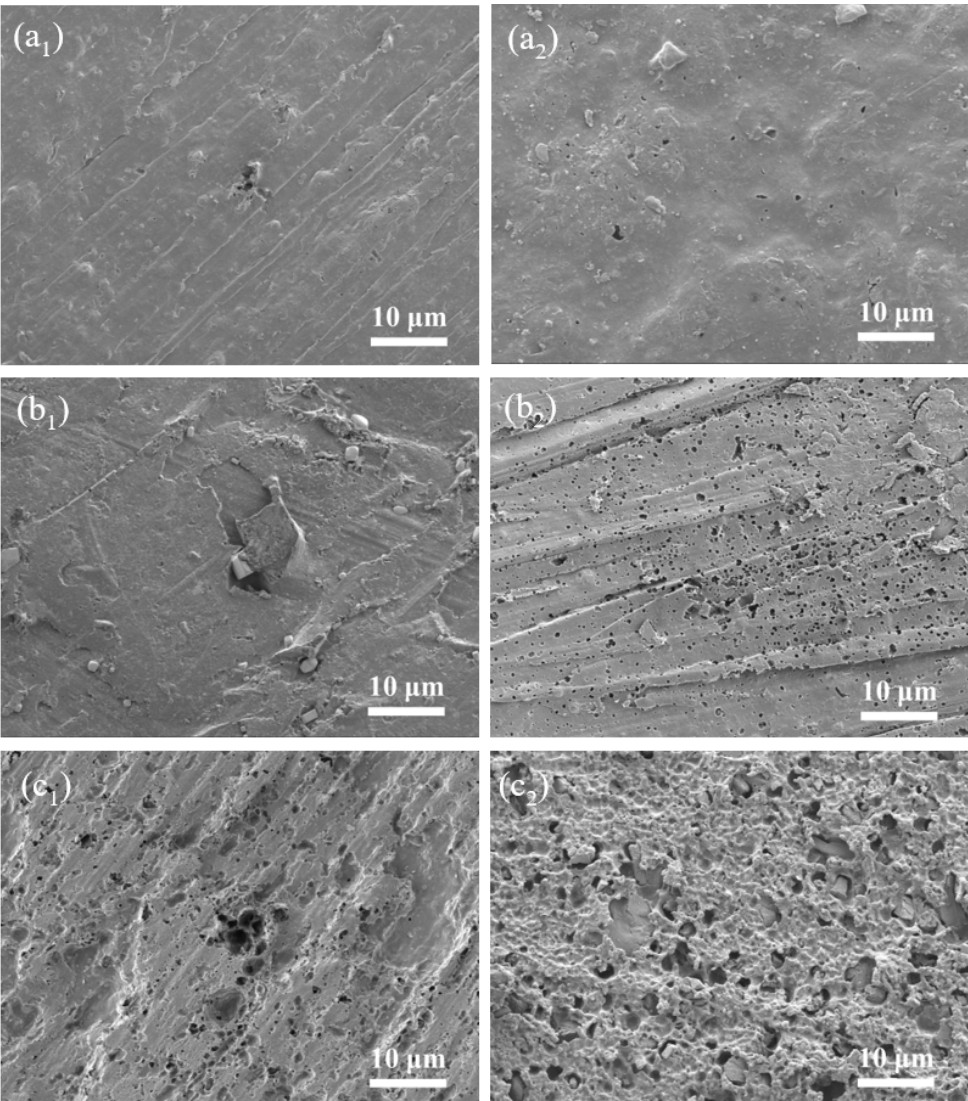

**Figure 10.** Coating morphology of contact surface with the metal matrix after immersion in static seawater (**a₁–c₁**) and flowing seawater (**a₂–c₂**) for 312 h: under OCP (**a₁,a₂**), under −850 mV $_{\text{vs. SCE}}$ (**b₁,b₂**) and under −1100 mV $_{\text{vs. SCE}}$ (**c₁,c₂**).

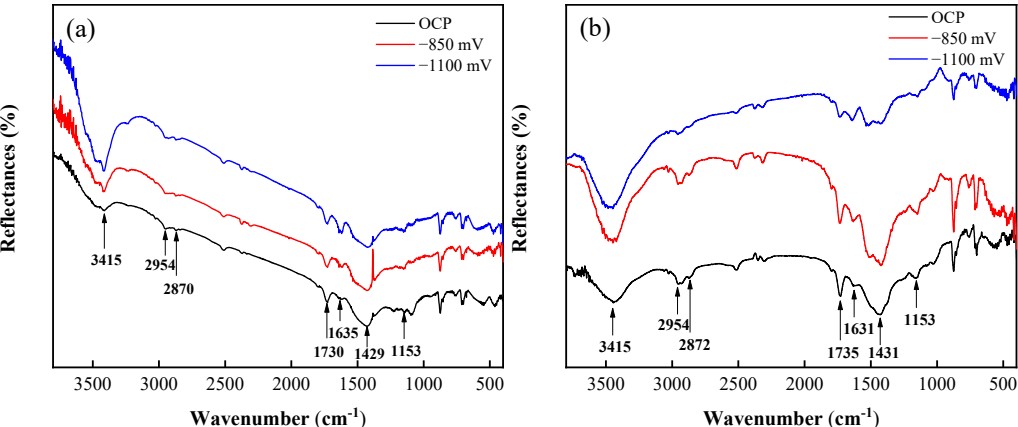

**Figure 11.** FT-IR spectra of organic coating in static (**a**) and flowing (**b**) seawater after 312 h of immersion.

The characteristic peak of the coating in flowing seawater did not change significantly. With the negative shifting of polarization potential, the intensities of the characteristic peaks at 3415 cm$^{-1}$ and 1631 cm$^{-1}$ increased significantly, and the characteristic peak representing the ester group at 1735 cm$^{-1}$ decreased. These phenomena were consistent with the results of immersion in static seawater, indicating that the hydrolysis of ester bonds in the alkyd resin was intensified. The above results indicated that the chemical structure changes of the coating were the same in the two seawater environments, and a more negative polarization potential could significantly affect the chemical structure of the coating.

## 4. Conclusions

(1). The SEM and EIS results showed that the cathodic reaction with the metal matrix, as well as the degradation of the organic coating, could be accelerated by cathodic polarization in both static and flowing seawater. This phenomenon became serious with the negative shifting of polarization potential; the impedance under −1100 mV was one order of magnitude lower than that under the OCP.

(2). Under the condition of OCP, the delamination area ratios in static and flowing seawater were 8% and 11%, respectively. The diffusion of corrosive particles into the coating could be accelerated by the flowing seawater, and the dissolved oxygen consumed by the cathodic reaction at the interface could be replenished in time, resulting in a faster electrochemical reaction speed, promoting the cathodic delamination of the coating.

(3). Under the same cathodic polarization potential, the delamination rate in flowing seawater was first higher than that in static seawater and then lower. The reason was that the anodic dissolution reaction of the metal matrix in flowing seawater enhanced with the increasing cathodic delamination area. Then, the cathodic polarization, cathodic reaction, and cathodic delamination were weakened accordingly.

(4). The FT-IR results showed that alcohols and carbonyl species were formed during the chemical degradation process of the coating in the two seawater environments. A more negative polarization potential could significantly affect the chemical structure of the coating.

**Author Contributions:** H.G.: Investigation, Writing-original draft, Writing-review & editing. K.Z.: Investigation, Software. Z.F.: Writing—review. C.L. (Chengjie Li): Investigation. J.X.: Formal analysis, Investigation. J.M.: Data curation, Supervision. X.Z.: Methodology, Formal analysis. X.W.: Formal analysis. K.X.: Investigation. C.L. (Chuanpeng Li): Data curation, Software, Supervision. J.L.: Funding acquisition, Formal analysis, Writing—review. All authors have read and agreed to the published version of the manuscript.

**Funding:** This work was financially supported by National Natural Science Foundation of China (Grant No. 51971192), Equipment Pre-research Field Fund (Grant No. 80904010503), Natural Science Foundation General Project of Shandong Provincial (Grant No. ZR2020ME132 & Grant No. ZR2022QE155).

**Institutional Review Board Statement:** Not applicable.

**Informed Consent Statement:** Not applicable.

**Data Availability Statement:** Not applicable.

**Conflicts of Interest:** The author declares no conflict of interest.

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
