# Peer review of "Corrosion Behavior of Alkyd-Resin-Coated Carbon Steel under Cathodic Polarization in Both Static and Flowing Seawater"

_coatings, doi:10.3390/coatings13071296_

Round 1
Reviewer 1 Report
The results displayed that both the diffusion of the corrosion particles and the anodic dissolution reaction of the metal could be significantly affected by the cathodic polarization and the flowing condition of the solution, which provided a possible approach to give an insight into the delamination of organic coating. Some comments given as follows.
1. Line 28, explain the criteria of effective protection intend by the authors.
2. Line 47, please paraphrase the phrase “coating/metal”.
3. Line 83, for the conducted method, there is a basis of standard? Provide it.
4. Line 100 of Figure 1, detail of fabrication should be given.
5. Line 133 for section 2.5, please more detail.
6. In the present form, actually nothing really novel. The current works appears to be a replication or modified literature according to the lack of novelty. The authors must extensively describe the novel in their work. This work should be rejected due to a serious concern.
7. Effort regarding minimize corrosion apart from using coating in transportation application, is also using composite materials, such as Carbon-fiber-reinforced polymers (CFRPs) as explained by Mughal et al. Please incorporated this explanation along with relevant reference as follows: https://doi.org/10.3390/pr11051540
Reviewer 2 Report
Thank the authors for submitting this exciting paper to Coatings Journal. Overall, I believe this paper is informative for the scientific community. However, some significant modifications must be applied to the article, essential for further consideration. Please see my detailed comments below. All the comments are mandatory. Please address all comments in detail to avoid delay in the peer-review process or possible unfavorable outcomes. Overall, the authors must verify the utilized electrical circuit models for simulating the EIS profiles, both statistically and physically. Without such an analysis, any discussion on the EIS profiles is baseless.
Comment (1): Please add more quantitative information about the coating corrosion performance to the abstract section.
Comment (2): What is the basis for the selection of amplitude of sinusoidal disturbance of 20 mV? Please provide supporting references.
Comment (3): Please specify the counter to the working electrode surface area.
Comment (4): To statistically verify the goodness-of-fit, please report the chi-square and sum of squares values in Table 2 and discuss the significance of those two parameters.
Comment (5): The Nyquist plots must be squared.
Comment (6): Please report the bode plots of the total impedance magnitude and the phase angle. Without those plots, it is impossible to comment on the robustness of the fits.
Comment (7): The authors utilized the Warburg impedance (diffusion) in the equivalent electrical circuit models. However, the authors did not discuss the physical interpretation of the Warburg impedance. Please elaborate on it.
Comment (8): In Tables 2 and 3, the symbol of the constant phase element is CPE. The C is the symbol of capacitance. Please revise.
Comment (9): Please discuss the physical interpretation of the CPE utilized in the equivalent electrical circuits.
Comment (10): It is essential to add a schematic of the electrochemical interface to clarify the physical interpretation of the equivalent electrical circuits.
A careful proof-read is recommended.
Reviewer 3 Report
This paper deals with corrosion behaviour of Alkyd Resin Coated Carbon Steel under Cathodic Polarization in both Static and Flowing Seawater. The topic is interesting and has application in metal structures especially pipeline steels. The introduction provides a clear overview of the topic with up to date references. The scope is well stated whereas the paper has clear structure. Overall, conclusions are based on results of this study. For this reason, it is recommended to be accepted after minor revision. Below the authors will find certain remarks that need to be addressed prior to publication.
Abstract: the methods used for examination must be mentioned in abstract.
Lines 76-78: Could you please mention the method (s) used for studying the induction cathodic delamination mechanism and development process of organic coating under different cathodic polarization potentials and seawaters environments?
Lines 78 -80: Could you please mention the methods used to characterize the degradation mechanism of coating?
Reviewer 4 Report
This study reports the effect of cathodic polarization on the corrosion behavior of alkyd resin coated in saline solution in both static and flowing conditions. The subject is interesting for the readers of the journal. Moreover, this also seems to be a revised version (going through the 4 rounds of review) of the manuscript. My only concern is the English. There are a lot of typos and grammatical errors. See the attached PDF of the manuscript with the highlighted errors. I have only reported the ones from the abstract and introduction as examples, but there are others in the other section.

There are a lot of typos and grammatical errors that need to be fixed.
Reviewer 5 Report
The authors studied the corrosion behavior of Alkyd resin coated carbon steel under cathodic polarization in static and flowing seawater. The manuscript is interesting and well-managed and can be published after minor revision:
1. Some typos and grammatical mistakes are present; please check it over all the manuscript.
2. The authors should emphasize the importance of their work.
3. Unit of Warburg is missing in tables 2 and 3
4. The resolution of the figures should be improved and increased.
5. The conclusion should contain results.
English language is not bad
Round 2
Reviewer 1 Report
Well done to the authors, some comments given in the revised form.
1. Line 130, the additional data that explain EIS measurement procedure is encouraged.
2. Line 140, please recheck the additional values added.
3. Line 141-142, for the software used ,please add company, city, and county information.
4. The authors need to explain potential further study of metals materials using computational simulation. It brings the advantages such as lower cost and faster results compared to experimental analysis as performed in the present stuy. Please include this explanation along with relevant reference as follows: https://jurnaltribologi.mytribos.org/v33/JT-33-31-38.pdf
-
Reviewer 2 Report
Thank the Authors for carefully addressing my comments.
Some minor modifications are suggested in the application of "the."
Round 3
Reviewer 1 Report
Great effort from authors, but some correction still needed.
1. Line 90, the authors state that the coating would characterise by SEM and FT-IR. Please explain the basic concept of SEM and FT-IR, where SEM is widely used to observe morphological structures sample surface at high magnification using a beam high energy electrons and FT-IR aims to characterize the functional groups in a sample. Please state this information and refer the relevant reference as follows: https://doi.org/10.12911/22998993/158564
2. Line 104, please explain more detail related ISO 15184-2012 for better understanding.
3. Line 112, in illustration please give the dimension of initial WBE to easier captured by the reader.
4. Line 113, for Table 3, please recheck the values.
5. Line 114, please clarify, where is the Q235 data obtained.
6. Line 120, for OCP, recommended presented in the table.
-
